# MorphTok: Morphologically Grounded Tokenization for Indic languages

**Maharaj Brahma** [1]   **NJ Karthika** [2]   **Atul Singh** [2]   **Devaraj Adiga** [3]   **Smruti Bhate** [3]   **Ganesh Ramakrishnan** [2 4]
**Rohit Saluja** [5 4]   **Maunendra Sankar Desarkar** [1 4]

## Abstract

Tokenization is a crucial step in NLP, especially with the rise of large language models (LLMs), impacting downstream performance, computational cost, and efficiency. Existing LLMs rely on the classical Byte-pair Encoding (BPE) algorithm for subword tokenization that greedily merges frequent character bigrams, often leading to segmentation that does not align with linguistically meaningful units. To address this, we propose morphology-aware segmentation as a pretokenization step before applying BPE. To facilitate morphology-aware segmentation, we create a novel dataset for Hindi and Marathi, incorporating sandhi splitting to enhance the subword tokenization. Experiments on downstream tasks show that morphologically grounded tokenization improves machine translation and language modeling performance. Additionally, to handle the dependent vowels common in syllable-based writing systems used by Indic languages, we propose Constrained BPE (CBPE), an extension to the standard BPE algorithm incorporating script-specific constraints. In particular, CBPE handles dependent vowels to form a cohesive unit with other characters instead of occurring as a single unit. Our results show that CBPE achieves a 1.68% reduction in fertility scores while maintaining comparable or improved downstream performance in machine translation and language modeling, offering a computationally efficient alternative to standard BPE. Moreover, to evaluate segmentation across different tokenization algorithms, we introduce a new human evaluation metric, *EvalTok*, enabling more human-grounded assessment.

[1]Department of CSE, IIT Hyderabad, India [2]Department of CSE, IIT Bombay, India [3]TIH, IIT Bombay, India [4]BharatGen Consortium [5]School of Computing and Electrical Engineering, IIT Mandi, India. Correspondence to: Maharaj Brahma <cs23resch01004@iith.ac.in>, NJ Karthika <karthika@cse.iitb.ac.in>.

*Proceedings of the ICML 2025 Tokenization Workshop (TokShop)*, Vancouver, Canada. PMLR 267, 2025. Copyright 2025 by the author(s).

| Word | BPE Segments | Morphologically Grounded Segments |
|---|---|---|
| खुलता | खु  लता | खुल  ता |
| उपजता | उप  जता | उपज  ता |
| कांडला | का  ◌ंड  **ला** | कांड  **ला** |
| गोलार्ध | **गोल**  ◌ार्  ध | **गोल**  अर्ध |

*Figure 1.* An example of segments generated by Byte Pair Encoding (BPE) compared with morphologically grounded segments. In this illustration, segments are separated by double space, and bold segments indicate correct segments from BPE with the ground truth.

## 1. Introduction

Tokenization forms the first step in any Natural Language Processing (NLP) pipeline. It is the process of dividing the text into smaller units, namely tokens, for further text processing. The tokens thus formed may be phrases, words, sub-words, or even characters, which form the smallest processing unit of the text, and hence, the quality of the tokens plays a crucial role in any NLP task. The most widely accepted and used tokenization method is Byte Pair Encoding (BPE) (Gage, 1994; Sennrich et al., 2016). BPE algorithm works by breaking a given text into individual characters (Unicode characters) or bytes and then building tokens by merging the most frequent bigrams iteratively. These merges are then stored in an ordered sequence. During tokenization, an input word is first split into individual characters. The learned merges are then applied sequentially, starting from the most frequent merges. BPE has been widely adopted in NLP due to its simplicity, effectiveness in handling OOV words, and its ability to control vocabulary size.

Despite its effectiveness, BPE operates greedily by picking frequent adjacent bigrams and merging them without considering linguistic structure. As a result, learned merges may violate the morpheme or word boundaries, leading to undesirable and linguistically incoherent segmentations. Figure 2 shows comparative examples of tokens generated by the BPE algorithm and the corresponding morphologically grounded tokens. For example, the word खुलता (khulatā[1],

---

[1]We follow the Roman transliteration scheme ISO 15919.

opens)[2] is formed by the verb root खुल (khula, open) and the suffix ता (tā), which BPE incorrectly tokenizes to खु (khu, -) and लता (latā, climber), where the component tokens do not preserve the meaning represented by the original word. This issue can become more pronounced in multilingual settings, where different languages exhibit distinct morphological patterns. To address this issue, we extend the concept of pre-tokenization, responsible for performing a morphologically grounded split based on a linguistically curated lookup table (see Section 3.1), as an additional step before tokenization.

To address the linguistic inconsistencies in subword tokenization, we introduce a novel approach to pre-tokenization, discussed in Section 3.1, which aims to align token segmentation with morpheme boundaries. Existing tokenization algorithms, such as BPE or Byte-based BPE, start with characters or bytes initialization. In the Latin script, letters are written sequentially from left to right. In contrast, the Devanagari script organizes symbols into syllabic units. Each syllable contains a single vowel at most, and whenever possible, syllables avoid ending in consonants. Due to character-level initialization, the dependent vowels are considered as a separate token. This leads to extra segmentation, not adhering to the written form. Inspired by this, we introduce a constraint during the initialization of the BPE algorithm. Ensuring dependent vowels do not form separate tokens, thus improving compression (see Section 3.2). Our key contributions are as follows:

- **Morphologically grounded pre-tokenization**: A linguistically motivated segmentation step that aligns subword units with morpheme boundaries, improving linguistic coherence over standard BPE.

- **Constrained BPE (CBPE)**: A simple extension to BPE that prevents dependent vowel diacritics from forming separate tokens, reducing token fragmentation in abugida scripts while maintaining comparable downstream performance.

- **EvalTok**: A human-centric evaluation metric that assesses morphological and semantic quality of tokenization. EvalTok complements automated metrics and enables qualitative comparison across tokenizers.

- **Indic segmentation dataset**: We release a curated morphological segmentation dataset for Hindi and Marathi (54k and 58k entries), supporting research in morphology-aware tokenization.

- **Comprehensive evaluation**: We benchmark our approach on machine translation and language modeling tasks. Our analysis highlights benefits beyond surface-level performance metrics, emphasizing linguistic fidelity.

---

[2]Format followed is word (roman transliteration, gloss)

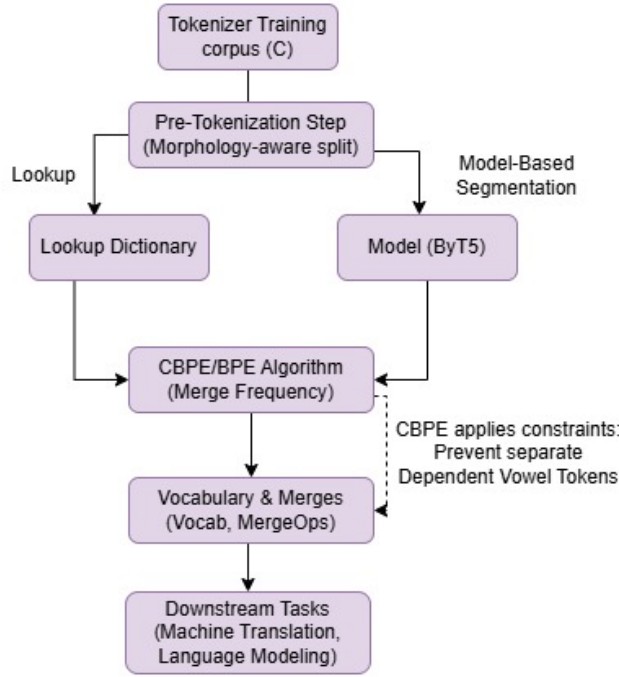

*Figure 2.* MorphTok tokenization pipeline. Tokenizer Training corpus is segmented using either a lookup dictionary or ByT5 model, followed by CBPE which applies script-specific constraints before generating vocabulary for downstream tasks.

## 2. Related Work

In the early years of NLP research, the most commonly used method of tokenization was splitting input text into space-separated words (white-space tokenizers) or characters. With the evolution of statistical and ML-based NLP in the late 1900s and early 2000, systems required a more evolved method of tokenization as well, such as n-gram-based, rule-based, and methods using finite-state automata. The advent of deep learning necessitated further sophisticated methods for tokenization. During this time, the tokenization method included statistical and probabilistic approaches. The most prominent and widely used tokenization that continues to be in use today, even with LLMs, is co-occurrence-based subword-level tokenizing methods like Byte Pair Encoding (Sennrich et al., 2016), Unigram (Kudo, 2018), Sentence Piece (Kudo & Richardson, 2018) and their variants. Some of the variants include prioritizing the merge of longest tokens (Lian et al., 2024), or starting the merge operations by splitting a word into longest subsequences matching vocabulary entries instead of splitting the word into single characters (Balde et al., 2024) in the traditional BPE method.

The unsupervised tokenization methods have obvious downsides, as frequency-based tokenization does not necessarily ensure correct morphological boundaries to form indepen-

dently meaningful tokens (Banerjee & Bhattacharyya, 2018). This issue is particularly prominent for Indian languages, as in many cases, combining tokens in Indian languages also leads to changes in characters at the word boundaries (sandhi), which cannot be captured by frequency-based tokenization methods. Recent literature includes works that factor in semi-supervision, as well as information related to the respective language's morphology. Bauwens & Delobelle (2024) identifies unnecessary BPE merges using a blame metric and removes the corresponding subwords from the vocabulary. However, such studies are limited to non-Indian languages.

## 3. Methodology

In this section, we describe our methodology. Section 3.1) outlines the pre-tokenization process, beginning with word and morphologically grounded segments dictionary and lookup-based approach in Section 3.1.1. We then present the model-driven pre-tokenization method in Section 3.1.2. In Section 3.2, we describe our method to handle dependent vowels.

### 3.1. Pre-Tokenization

Most of the popularly used tokenization algorithms follow greedy merging approaches based on the frequency of bi-grams. Such methods of tokenization do not guarantee morphologically grounded subword tokens, especially in cases of morphologically rich languages (Nzeyimana & Rubungo, 2022; Arnett & Bergen, 2025). Most of the Indian languages face the risk of forming lossy subwords by following such simple frequency-based methods alone for tokenization. For example, the word सूर्योदय (sūryōdaya,sunrise) is formed from the 2 components {सूर्य (sūrya, sun), उदय (udaya, rise)} following sandhi rules. The best possible outcome of tokenization of this word by BPE would be {सूर्य, ोदय} {(sūrya, sun), (ōdaya,-)} or {सूर्यो, दय} {(sūryō, sun), (daya, mercy)}. In both these cases, the component splits do not preserve the correct meaningfulness of the subwords. Hence, we require a more linguistically grounded process for tokenization.

Two common types of word segmentation datasets for Indian languages are: (a) segmentation based on sandhi, which yields semantically and linguistically correct sub-word segments. Such segmentation may involve changes at the sub-word boundaries, (b) lossless word-segmentation method, where sub-words do not have any character changes, and their concatenation yields the original word. In this case, the sub-words may not always be meaningful by themselves.

### 3.1.1. Lookup Based

We create a word segmentation dataset for two languages, Hindi and Marathi, with the aid of language experts. The

---

**Algorithm 1** Morphologically Grounded Tokenization

**Require:** Training Corpus $\mathcal{C}$; No. of Merges $\mathcal{K}$; Pre-tokenization Type $\mathcal{T}$ ($\mathcal{T} \in \{$Model, Lookup$\}$); Lookup $\mathcal{L}$ (Pairs of Word $\mathcal{W}$ and Segments $\mathcal{S}$)
**Ensure:** Vocabulary $\mathcal{V}$, Merges $\mathcal{M}$
1: $\mathcal{C}' \leftarrow$ PreTokenize($\mathcal{C}, \mathcal{T}$)
2: $\mathcal{V}, \mathcal{M} \leftarrow$ BPE($\mathcal{C}', \mathcal{K}$) {Learn merges using BPE}
3: **function** PreTokenize ($\mathcal{C}, \mathcal{T}$)
4:   **if** $\mathcal{T} =$ Model **then**
5:     $\mathcal{U} \leftarrow$ ExtractUniqueWords($\mathcal{C}$)
6:     $\mathcal{D} \leftarrow$ WordSegmentationModel($\mathcal{U}$)
7:     $\mathcal{C}' \leftarrow$ ApplySegments($\mathcal{C}, \mathcal{D}$)
8:   **else**
9:     $\mathcal{C}' \leftarrow$ ApplyLookup($\mathcal{C}, \mathcal{L}$)
10:   **end if**
11:   **return** $\mathcal{C}'$
12: **end function**

---

*Table 1.* Lookup dataset statistics for Hindi and Marathi

| Language | Total word-segment pairs |
|----------|--------------------------|
| Hindi | 54,395 |
| Marathi | 58,333 |

methods used to create the dataset are as follows: (a) automatic generation. With the aid of language experts, we list common affixes for nouns and verbs separately and automatically generate all possible combinations of stems with the corresponding affixes. (b) We use an existing word segmenter model (Bhatt et al., 2024) to generate the initial word splits, which are further post-edited by language experts to obtain morphologically and semantically correct word segments.

Each entry in the lookup table $\mathcal{L}$ maps a word $\mathcal{W}$ to its morphologically grounded segments $\mathcal{S}$. During the pre-tokenization stage, every occurrence of $\mathcal{W}$ in the tokenization training corpus $\mathcal{C}$ is replaced with the corresponding segments $\mathcal{S}$. We then apply standard BPE algorithm to the resulting pre-tokenized corpus.

### 3.1.2. Model-driven Word-segmentation

The human-curated dictionary lookups are limited in both size and coverage. To address this, we explore the potential usage of model-based segmentation methods to enhance lookup coverage. To train the model to recognize cases where no segmentation is required, we treat the first split from the lookup table as a word. For both Hindi and Marathi, we lookup table is divided into training, validation, and test sets. We initially experimented with character-level Bi-LSTM models. However, these models struggle to capture sandhi-based patterns effectively. To improve performance, we fine-tune the pre-trained mT5 model (Xue et al., 2021), leveraging its multilingual pretraining capabilities. However, we hypothesize that the presence of a tokenizer

in pre-trained models may negatively impact segmentation performance. To mitigate this issue, we further fine-tune the byte-level tokenization-free ByT5 model (Xue et al., 2022), which yields improved segmentation performance. A detailed analysis of model selection and performance comparison is provided in Section 5.1.

In model-driven word segmentation, we begin by extracting the set of unique words $\mathcal{U}$ from the tokenization training corpus $\mathcal{C}$. These words are then passed through a word segmentation model in our case a fine-tuned ByT5 model, which produces a segmented dictionary $\mathcal{D}$. The output is subsequently filtered to obtain a refined dictionary $\mathcal{D}'$ containing high-confidence segmentations. Here, we employ a rule-based filtering strategy. Finally, we generate the pre-tokenized corpus by replacing each word in the original corpus that appears in the refined dictionary with its corresponding segments. The formal algorithm for the morphological grounded tokenization are presented in Algorithm 1.

### 3.2. Constraining Dependent Vowels

Linguistic diversity of written scripts across the world poses significant challenges for the tokenization process, particularly in languages that follow the abugida[3] writing system. Unlike alphabetic scripts, where vowels and consonants are treated as independent units, abugida scripts follow a consonant-vowel system. Especially in Indian languages, the Devanagari script has a set of dependent and independent vowels. The dependent vowels are represented in the form of diacritics. Existing statistical tokenization algorithms, such as BPE, are primarily designed for alphabetic scripts, operating at the level of Unicode characters or byte-based methods starting from bytes encoding[4] to learn the merges. Consequently, BPE frequently learns merges that are linguistically obvious. We empirically find that approximately 5% of merges in a 32k BPE merges are dedicated to combining characters with dependent vowels. This effect is even more pronounced with smaller merges sizes such as 8k and 16k, as shown in Table 2.

*Table 2.* Obvious merges in the BPE algorithm for 8k, 16k, and 32k merge sizes, calculated as the number of merges where the second token is a dependent vowel in the Devanagari script.

| # of merges ($\mathcal{K}$) | # of obvious merges |
| --- | --- |
| 8k | 861 (10.76%) |
| 16k | 1203 (7.52%) |
| 32k | 1739 (5.43%) |

To address this issue, we introduce Constrained BPE (CBPE), a simple extension to the BPE algorithm that ex-

[3]https://en.wikipedia.org/wiki/Abugida
[4]UTF-8 based

plicitly preserves dependent vowels during tokenization. In standard BPE, the algorithm initializes with individual characters or Unicode. In contrast, CBPE modifies this initialization step by attaching dependent vowels to their preceding Unicode characters, as illustrated in Figure 3. This ensures that the consonant-vowel units remain intact, preserving linguistic coherence. Once initialized, CBPE follows the standard BPE merge learning procedure i.e. selecting merges that have high frequency. The merges learned using CBPE ensure obvious merges are reduced. During tokenization, CBPE applies similar constraints on dependent vowels and consecutively applies merges similar to the BPE algorithm. Hence, CBPE ensures that the dependent vowels do not form separate tokens or avoid tokens starting with dependent vowels during the tokenization process. A formal description of the algorithm is presented in Algorithm 2. For pre-tokenization followed by CBPE, we replace BPE in line 2 of Algorithm 1 with the CBPE algorithm.

---

**Algorithm 2** CBPE (Constrained BPE) Algorithm

**Require: Input:** Training Corpus $\mathcal{C}$; Number of Merges $\mathcal{K}$
**Ensure: Output:** Vocabulary $\mathcal{V}$, Merges $\mathcal{M}$
1: $\mathcal{V} \leftarrow \emptyset, \mathcal{M} \leftarrow \emptyset$
2: Initialize vocabulary with dependent vowels attached to preceding Unicode characters
3: **while** $|\mathcal{V}| < \mathcal{K}$ **do**
4:     $(t_l, t_r) \leftarrow$ Select the most frequent bigram pair in $\mathcal{C}$
5:     $\mathcal{V} \leftarrow \mathcal{V} \cup \{t_l t_r\}$
6:     $\mathcal{M} \leftarrow \mathcal{M} \cup \{(t_l, t_r)\}$
7:     Replace all occurrences of $(t_l, t_r)$ with $t_l t_r$ in $\mathcal{C}$
8: **end while**

---

The effects of our proposed methods, including lookup-based pre-tokenization and constrained BPE, are empirically evaluated in the next Section 4 (Experiments), focusing on their impact on machine translation and language modeling.

## 4. Experiments

In this section, we describe our experimental setup to answer the following set of questions: (a) Does lexically grounded segmentation combined with a statistical tokenization algorithm improve performance in machine translation and language modeling tasks? (b) Does model-driven lookup creation have better performance than a human-created lookup? (c) Does constraining dependent vowels from forming a separate token have better or equal performance to that of BPE?

### 4.1. Segmentation Encoding

To distinguish between the segmentations produced by the lookup and BPE methods, we utilize two distinct segmentation markers. The ** symbol is employed for both lookup and model-based segmentations, while the @@ symbol

| Word | BPE Initialization | CBPE Initialization |
|------|--------------------|--------------------|
| क़लम | क__◌.__ल__म | क़__ल__म |
| पढ़ाई | प__ढ__◌.__◌ा__ई | प__ढ़ा__ई |
| कार्यालय | क__◌ा__र__◌्__य__◌ा__ल__य | का__र्__या__ल__य |

*Figure 3.* BPE and CBPE initialization

specifically denotes segmentations generated by the BPE algorithm across all experiments.

## 4.2. Tokenizer Evaluation

Intrinsic evaluation of tokenizers remains challenging as there are no standard intrinsic metrics that correlate well with downstream performance. The community commonly relies on the fertility metric (Rust et al., 2021)—the average number of subwords produced per tokenized word. A lower fertility score generally indicates more efficient tokenization with fewer subword fragments per word. However, in morphologically rich languages, higher fertility scores may be necessary to model and capture linguistic structures appropriately.

To address this, we adopt a multi-faceted evaluation strategy. First, we use downstream task performance (machine translation and language modeling) to assess the practical utility of each tokenization method. Second, we introduce a new metric, *EvalTok*: Human Post-hoc Evaluation of Tokenization, to capture the semantic and morphological adequacy of subword segmentations—factors often overlooked by automatic metrics.

We sample 100 words from a test set and perform a human evaluation on the segmentation quality of BPE and Lookup-based pre-tokenization. We define a metric on a scale of 1–4 to rate the quality of segmentation. This two-pronged evaluation is motivated by the need to balance structural efficiency (via fertility and downstream performance) with linguistic coherence (via human evaluation). Automatic metrics like fertility are informative for measuring fragmentation, while EvalTok provides human-grounded validation of morphological correctness.

The scoring rubrics followed by the language experts are as follows:

- **1:** None of the tokens are morphologically correct and neither preserve the semantics of the original word.
  Example: If the word खुलता = खुल + ता (khulatā = khula + tā) is tokenized to खु (khu,-) and लता (latā, climber), both the tokens are incorrect and do not preserve the correct semantic meaning of the original word.
  *Note: Here, the word लता is independently a semantically correct word meaning climber, but in the context*

*of the original word, it is incorrect.*
- **2:** > 50% of the tokens do not preserve the morphology or semantics in the context of the original word.
  Example: गोलार्ध (gōlārdha, hemisphere) = गोल (gōla, sphere) @@ ◌ार् (ār, -) @@ ध (dha, -)
  Here, the first token गोल is correct while the second and third are incorrect tokens (both morphologically and semantically)
- **3:** >= 50% of the tokens are either morphologically or semantically correct.
  Example: The word चित्रा (citrā) is ideally not to be tokenized further. But in case the word is tokenized to चित्र (citra) @@ ◌ा (ā), the token चित्र do preserve the meaning in the context of the original word and hence scored positively.
- **4:** All the tokens are morphologically and semantically correct. The words that aren't tokenized are also given the high score.
  Example: छायाचित्र (chāyācitra, photograph) = छाया (chāyā, shadow) @@ चित्र (citra, picture). Here both the tokens are morphologically and semantically correct.

Since the fertility metric does not fully reflect the linguistic validity of token splits, we evaluate the tokenization performance of the Lookup + BPE algorithm using both downstream task performance and human evaluation via EvalTok. For CBPE, we report fertility, downstream task performance, and human evaluation to assess its effectiveness in reducing undesirable subword boundaries, particularly for dependent vowels in Indic scripts.

In the next Section 4.3, we present the implementation details. Subsequently, in Section 5.1, we present a more detailed discussion of CBPE's impact on fertility reduction and downstream performance.

## 4.3. Implementation Details

### 4.3.1. Model-driven Word segmentation

We performed our experiments using the Huggingface Transformers library[5]. We evaluate the model performance using Exact Match (EM), Precision (P), Recall (R), and F1 scores (Bhatt et al., 2024). We observe that finetuning on large

---

[5] https://github.com/huggingface/transformers

models can overfit, so we restrict the experiment to only small (300M) and base (580M) parameter models for mT5 and the base model for ByT5. The results for mT5 and ByT5 fine-tuning are provided in Appendix C. Hyper-parameters details are presented in Appendix E.

### 4.3.2. Downstream Task

**Machine Translation:** We perform machine translation experiments for Hindi to Marathi and Marathi to Hindi language directions for 16k and 32k merges. We use a standard transformer model (Vaswani et al., 2017) with 6 encoder and decoder layers. The model is trained for a maximum of 100k updates using the Adam (Kingma & Ba, 2014) optimizer with $\beta_1 = 0.9$ and $\beta_2 = 0.98$. We use a dropout of 0.2 and apply gradient clipping with a norm of 1.0. We set a learning rate of $5 \times 10^{-4}$. Before training, we preprocessed and normalized the data using IndicNLP[6] library. We perform our experiments using fairseq[7] library.

We evaluate the translation performance using both automatic and human evaluation metrics. In automatic metrics, we employ lexical-based metrics such as BLEU (Papineni et al., 2002), and chrF (Popović, 2015), along with model-based metrics like COMET (Rei et al., 2020; 2022)[8]. For human evaluation, we assess 100 randomly sampled translation outputs using the widely-used XSTS (Licht et al., 2022) metric, rated on a scale from 1 to 5. We report our results on the In22-Gen (Gala et al., 2023) test set. To ensure control over our experiments, we apply lookup and model-based pre-tokenization only on the source text. Experiments were conducted on four NVIDIA H100 80 GB GPUs.

**Language Modeling:** We train a 355M-parameter language model based on the GPT-2 Medium architecture (Radford et al., 2019), using various tokenization strategies. In particular, we evaluate our proposed lexically grounded tokenization approach, which combines a linguistically curated lookup-based pre-tokenization step with Byte Pair Encoding (BPE) using 32k merge operations. For Hindi and Marathi, the model is trained on 1 billion words drawn from the Sangraha corpus (Khan et al., 2024). We use the Fairseq framework to conduct all language modeling experiments and assess model performance using perplexity and cross-entropy loss on a held-out set of 500 sentences. Detailed training configurations are provided in Appendix E.

## 5. Results and Discussions

In this section, we discuss our results and observations. Machine translation scores on automatic metrics for BPE,

---

[6] https://github.com/anoopkunchukuttan/indic_nlp_library

[7] https://github.com/facebookresearch/fairseq

[8] We use reference-free wmt22-comet-da model

*Table 3.* Perplexity and loss metrics for the Hindi and Marathi languages on the language modeling task. Results are reported after training for 7 epochs.

| Tokenization Algorithm | Hindi | | Marathi | |
|---|---|---|---|---|
| | PPL | Loss | PPL | Loss |
| BPE | 350.00 | 8.45 | 107.45 | 6.748 |
| Lookup + BPE | **225.00** | **7.81** | **97.78** | **6.611** |
| CBPE | 240.00 | 7.68 | 113.36 | 6.825 |
| Lookup + CBPE | **151.00** | **7.24** | **99.83** | **6.641** |

`Lookup + BPE`, `Model WS+BPE`, `CBPE`, `Lookup + CBPE`, and `Model WS+CBPE` are presented in Table 4.

### 5.1. Quantitative Evaluation

**Morphologically Grounded Tokenizer vs. BPE:** In downstream machine translation tasks for Hindi to Marathi and Marathi to Hindi, we observe that lexical grounded pre-tokenization (`Lookup + BPE`) followed by BPE consistently yields a higher COMET score than that of BPE for 16k and 32k merges except for Marathi to Hindi direction with 32k merges, where both tokenization methods achieve similar COMET scores. In terms of chrF2 scores, for Hindi to Marathi, we see an improvement of +2.2 for 32k merges compared to BPE. For the Marathi to Hindi, we observe a minor improvement of +0.9 for 16k merges.

In the language modeling task, the `Lookup + BPE` configuration consistently outperforms standard BPE, achieving lower perplexity and loss values. Similarly, `Lookup + CBPE` yields substantial improvements over CBPE, indicating that incorporating linguistically informed pre-tokenization significantly enhances model performance. These findings underscore the value of lexically grounded tokenization in facilitating more effective subword segmentation and representation learning.

It is important to note that perplexity scores between BPE and CBPE-based models are not directly comparable, as they are trained with different vocabulary structures. Nevertheless, within each tokenization family, the inclusion of a lookup-based pre-tokenization step results in clear and consistent gains. The detailed results are summarized in Table 3.

**BPE vs. CBPE:** We observe a reduction in fertility scores for CBPE compared to BPE for 8k, 16k, and 32k merge operations, indicating the effectiveness of constraining dependent vowels during the vocabulary creation process of BPE. Notably, vocab with 8k merges showed a difference of 0.021 for Hindi, suggesting that CBPE is more effective for smaller vocabulary sizes. Fertility scores of Hindi and Marathi for 8k, 16k, and 32k merges for both BPE and CBPE on the In22-Gen are shown in Table 5.

*Table 4.* Machine Translation results on In22-Gen. chrF2 and COMET scores are reported for **Hindi to Marathi** and **Marathi to Hindi** translation.

| | Hindi → Marathi | | | | Marathi → Hindi | | | |
| | 16k | | 32k | | 16k | | 32k | |
| | chrF2 (↑) | COMET (↑) | chrF2 (↑) | COMET (↑) | chrF2 (↑) | COMET (↑) | chrF2 (↑) | COMET (↑) |
|---|---|---|---|---|---|---|---|---|
| **BPE** | 37.7 | 0.6428 | 35.2 | 0.6155 | 37.0 | 0.6035 | **36.8** | 0.5962 |
| **Lookup + BPE** | 36.5 | 0.6454 | 36.1 | 0.6301 | 37.9 | 0.6115 | 36.3 | 0.5962 |
| **Model WS + BPE** | 37.8 | 0.6433 | 36.1 | 0.6142 | 37.9 | 0.6072 | 36.3 | 0.5853 |
| **CBPE** | 37.3 | **0.6448** | **36.7** | **0.6274** | 38.4 | 0.6151 | **37.6** | **0.5954** |
| **Lookup + CBPE** | 37.1 | 0.6395 | **36.7** | 0.6261 | 38.4 | 0.6232 | 36.2 | 0.5946 |
| **Model WS + CBPE** | 37.6 | 0.6380 | 36.0 | 0.5144 | 37.0 | 0.5991 | 36.3 | 0.5788 |

*Table 5.* Fertility scores on In22-Gen for Hindi and Marathi

| | Hindi | | | Marathi | | |
| Algorithm | 8k | 16k | 32k | 8k | 16k | 32k |
|---|---|---|---|---|---|---|
| BPE | 1.2708 | 1.1612 | 1.0953 | 2.0952 | 1.8858 | 1.7240 |
| CBPE | **1.2495** | 1.1566 | **1.0925** | **2.0082** | **1.8174** | **1.6633** |

*Table 6.* Human evaluation of MT predictions for various tokenization settings for vocabulary size of 32k

| Source → Target | BPE | Lookup + BPE | Model WS + BPE |
|---|---|---|---|
| HIN → MAR | 1.98 | **2.06** | 1.94 |
| MAR → HIN | **2.85** | 2.81 | 2.80 |

For machine translation, CBPE yields higher COMET scores than BPE for Hindi to Marathi at 16k and 32k merges and for Marathi to Hindi at 16k merges. At 32k merges for Marathi to Hindi, the COMET scores of BPE and CBPE are comparable. In terms of chrF2 scores, we observe a gain of +1.4 for Marathi to Hindi translation for 16k merges compared to BPE. In the Hindi to Marathi direction, we observe a gain of +1.5 chrF2 for 32k merges.

Overall, our findings suggest that tokenization with constraining dependent vowels helps reduce fertility while maintaining comparable performance to BPE. In some cases, CBPE also leads to improved COMET and chrF2 scores.

**Lookup vs. Model-driven segmentation:** We observe that Lookup-based segmentation consistently performs better than Model-based segmentation in terms of COMET scores. This suggests that (a) linguistically grounded segmentation may not be necessary for all words, and (b) model-driven segmentation may introduce noise, requiring further verification through human evaluation.

### 5.2. Post-Hoc Human Evaluation

For a comprehensive assessment of tokenization quality, we employ the EvalTok metric, detailed in Section 4.2, which quantifies morphological correctness and semantic coherence in segmented tokens

#### 5.2.1. Human Evaluation of MT Results

Commonly used metric for the evaluation of MT results is the BLEU score. BLEU is infamously ignorant of the meaningfulness of the output and is highly dependent on the literalness of the reference translations. Hence, BLEU is not completely reliable, especially for morphologically rich languages, which often yield low scores for the said reasons. Therefore, we use the XSTS metric, as proposed

by Licht et al. (2022), as a method of post-hoc intrinsic (qualitative) evaluation by language experts. We randomly selected 100 sentences subjected to translation under the 3 tokenization settings *viz.* S1: default BPE tokenization, S2: pre-tokenization with lookup followed by BPE and S3: pre-tokenization with our segmenter model (Model WS), followed by BPE. Language experts[9] followed the XSTS metric to score the target predictions from all 3 tokenization settings.

Table 6 shows the human evaluation results of the MT output, using the XSTS metric for the three tokenization settings: S1, S2, and S3, as discussed above. The evaluation shows that the translation quality is better with setting 2 for Hindi → Marathi, with an increase in score of 0.8. The score is 0.4 lesser for S2 compared to S1 for Marathi → Hindi. The score with the setting S3 is slightly lower in both cases, which can be attributed to the possible errors from the segmentation model, yet it is promising to note that the values are not significantly lower than their counterparts.

#### 5.2.2. Human Evaluation of Tokenization

To analyze the quality of tokenization with BPE versus our method of pre-tokenization + BPE, we propose a new metric *EvalTok*, as described in Section 4.2. We randomly chose 100 words and their respective tokenized outputs in the two settings: (a) default BPE and (b) pre-tokenization + BPE[10] The language experts scored the tokenization based on the EvalTok metric as described in Section 4.2. The average score is 2.56 for setting (a) and 3.16 for setting (b). The results are consistent with our assumption that a morpholog-

---

[9]The experts assigned to the task have native/advanced level proficiency in both source and target languages.

[10]We chose the words only from the set of words that underwent the pre-tokenization step for better comparison.

*Table 7.* Number of the dependent vowels as a separate token for various LLMs tokenizers. Here, **Indic models** are LLMs trained specifically for Indian languages. **DV** represents **Dependent Vowels** of the Devanagari script.

| Models | Indic Model | DV count as separate token |
|---|---|---|
| LLAMA-3.1-8B | N | 12330 |
| GEMMA-2-2B | N | 2157 |
| NANDA | Y | 454 |
| SARVAM-1 | Y | 325 |
| CBPE | - | 0 |

*Table 8.* Comparison of Tokenization Algorithms using Rényi's efficiency and chrF2 score for **Marathi → Hindi** Machine Translation task.

| Tokenization algorithm | Rényi's efficiency | chrF2 score |
|---|---|---|
| Vocabulary size: 32k | | |
| BPE | 0.356 | **36.8** |
| Lookup + BPE | **0.372** | 36.2 |
| Vocabulary size: 16k | | |
| BPE | 0.393 | 37.0 |
| Lookup + BPE | **0.407** | **37.6** |

ically aware pre-tokenization will lead to better quality tokens. Sample human evaluation scores for BPE and Lookup + BPE using the EvalTok metric are shown in Figure 7.

## 6. Further Analyses

In this section, we present a detailed analysis of our approaches across different aspects. Specifically, we examine (a) dependent vowels in existing LLM tokenizers (Section 6.1), (b) lookup pre-tokenization and constraining in multilingual setup (Section 6.2), (c) downstream performance correlation with Rényi's efficiency (Section 6.3), and (d) word length and segmentation size (Section 6.4).

### 6.1. Dependent Vowels in Existing LLM Tokenizers

We quantify the dependent vowels of the Devanagari script appearing as a single token in existing tokenizers of popular multilingual LLMs: LLAMA-3.1.8B (Grattafiori et al., 2024), GEMMA-2-2B (Team et al., 2024), and LLMs trained focused on Indian languages such as SARVAM-1 (SarvamAI, 2024) and NANDA (Choudhury et al., 2024). We use the IN22-Gen Hindi benchmark corpus, consisting of 1024 sentences, particularly for each sentence, and we count the number of times dependent vowels are used as a separate token.

We observe that popular multilingual LLM tokenizers such as LLAMA-3.1-8B and GEMMA-2-2B trained with traditional statistical tokenization algorithms have high counts. Similarly, models that are explicitly trained on Indian language data also have a significant count. Table 7 shows the total counts for various tokenizers. In contrast, CBPE have zero dependent vowels as a separate token.

### 6.2. Multilingual (1 to M) translation

To further study the effectiveness of pre-tokenization with lookup and constrained BPE on a multilingual machine translation setup. We select 6 target languages: Dogri (doi), Konkani (gom), Maithili (mai), Marathi (mar), Nepali (npi), and Sanskrit (san), belonging to the same language family and similar script as the source language. Recall that the lookup-based pre-tokenization used in our multilingual

translation experiments is described in detail in Section 3.1.1, where we outline the dictionary construction process. We find that in multilingual settings, BPE has slightly better scores than Lookup + BPE. This suggests that applying lookup-based pre-tokenization only to the source language might not necessarily facilitate cross-lingual transfer. The results are reported in Table 16.

### 6.3. MT results correlation with Rényi's efficiency

Recent work on tokenizer evaluation: Rényi's efficiency (Zouhar et al., 2023) utilizes an information theory framework to measure the tokenization quality intrinsically to show a significant correlation with BLEU metric for English-German MT. Rényi's efficiency measures the ratio of the unigram entropy of the tokenized text to the maximum possible entropy given the vocabulary size.

*Table 9.* Comparison of Tokenization Algorithms using Rényi's efficiency and chrF2 score for **Hindi → Marathi** machine translation task.

| Tokenization algorithm | Rényi's efficiency | chrF2 score |
|---|---|---|
| Vocabulary size: 32k | | |
| BPE | 0.376 | 35.2 |
| Lookup + BPE | **0.378** | **37.4** |
| Vocabulary size: 16k | | |
| BPE | 0.408 | **37.7** |
| Lookup + BPE | **0.410** | 36.5 |

We analyze the correlation between chrF2 scores and Rényi's efficiency[11] on BPE and Lookup + BPE tokenization methods for both Hindi→Marathi and Marathi→Hindi translation. We compute Rényi's Efficiency on MT training data and set $\alpha = 2.5$. The results for Hindi to Marathi and Marathi to Hindi are shown in Table 9 and Table 8, respectively. For the Hindi → Marathi translation, we observe a positive correlation between Rényi's Efficiency and chrF2 for 32k vocabulary but a negative correlation for 16k. Conversely, in Marathi→Hindi, we observe a positive correla-

---

[11]We use https://github.com/zouharvi/tokenization-scorer to compute Rényi's efficiency.

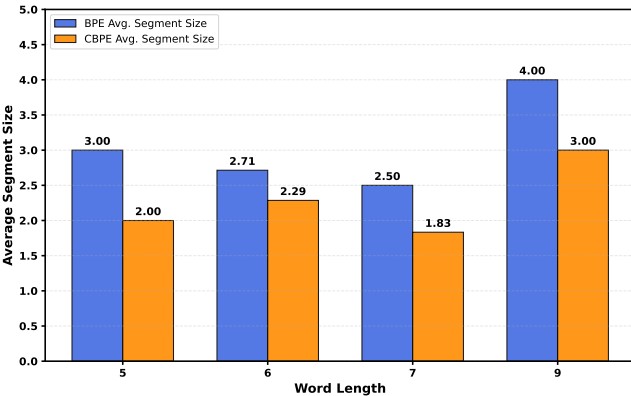

*Figure 4.* Comparison of Average Segment size for varying word length

tion for 16k vocabulary but a negative correlation for 32k. This suggests that the relationship between Rényi's Efficiency and translation quality depends on vocabulary size and translation direction. Our findings indicate that Rényi's efficiency is not always a reliable indicator of tokenization quality in machine translation, which is in line with observations made by (Libovický & Helcl, 2024). Further investigation is required to understand its variability across language directions and vocabulary size.

### 6.4. Word length and Segment size

We randomly sample 395 words with varying lengths and apply BPE and CBPE on merges learned for 32k merge operations. Then, we count the segment size with space separation. We exclude words that have the same segment size. On the remaining words, we compute the average segment size for BPE and CBPE for varying word lengths. We observe that, on average, CBPE has a smaller segment size than BPE, suggesting its effectiveness. Figure 4 shows the average segment size for BPE and CBPE groups according to word length.

## 7. Conclusion & Future Works

We introduced a new dataset for Hindi and Marathi to support a novel lookup-based pretokenization method followed by BPE, aiming to improve tokenization for low-resource languages. Our approach outperformed standard BPE in both machine translation and language modeling tasks and received higher scores in human evaluations. We also proposed a new human evaluation metric to better assess tokenization quality. By incorporating constraints on dependent vowels, our method (CBPE) effectively addressed common tokenization issues in syllabic writing scripts, reducing fertility without compromising model performance. Given its foundation in morphological and script-based features, the

method is extendable to other Indo-Aryan and Dravidian languages. Future work will explore multilingual extensions and the method's impact on larger language models.

## Impact Statement

This paper proposes incorporating linguistically grounded segmentation during the pre-tokenization stage compared to the statistically based tokenization algorithm. Also, we propose a method to respect the script-specific properties of Indic writing systems. These contributions have the potential to contribute to morphologically rich languages and more efficient training of large language models.

## Acknowledgements

We acknowledge BharatGen (https://bharatgen.com/) for providing resources and support to conduct the research. We are grateful to the data curators and human evaluators for dataset construction and translation quality assessment. Furthermore, we appreciate the insightful feedback and constructive suggestions from the anonymous reviewers, which helped improve the quality of this work.

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

## Appendix

### A. Lookup Data

Table 10 shows the sample entries in our dataset for Hindi. We are covering word splits from both internal Sandhi (leading to stem/root and affixes split) and external Sandhi (leading to multi-word split).

*Table 10.* Examples from the Lookup Hindi Data

| Word | Split 1 | Split 2 | Split 3 |
|------|---------|---------|---------|
| विद्यालय | विद्या | आलय | |
| उठता | उठ | ता | |
| उतारना | उतार | ना | |
| कराकर | करा | कर | |
| हडबडाना | हड | बडा | ना |
| कार्यालय | कार्य | आलय | |
| जगदम्बा | जगत् | अम्बा | |

### B. Hyperparameters & Dataset

The hyperparameters for language modeling experiments and model-based word segmentation are shown in Table 11 and 12, respectively. The dataset details for Multilingual analysis are shown in Table 13.

*Table 11.* Hyperparameter for Language Modeling

| Hyperparameter | Value |
|----------------|-------|
| Architecture | transformer_lm_gpt2_medium |
| Share Decoder Input-Output Embed | True |
| Dropout | 0.1 |
| Optimizer | Adam |
| Adam Betas | (0.9, 0.98) |
| Weight Decay | 0.01 |
| Clip Norm | 0.0 |
| Learning Rate | 0.0005 |
| LR Scheduler | inverse_sqrt |
| Warmup Updates | 4000 |
| Warmup Init LR | $1 \times 10^{-7}$ |
| Tokens per Sample | 16 |
| Max Tokens | 64 |
| Update Frequency | 16 |
| FP16 (Mixed Precision) | True |
| Max Updates | 500000 |

### C. Word Segmentation

Table 14 shows the word segmentation performance of various models.

### D. BLEU scores

The BLEU scores for Hindi → Marathi and Marathi → Hindi machine translation tasks are shown in Table 15.

*Table 12.* Hyperparameter for Model-based word segmentation

| Hyperparameter | Value |
|---|---|
| num_train_epochs | 30 |
| per_device_train_batch_size | 16 |
| per_device_eval_batch_size | 4 |
| logging_steps | 1000 |
| save_steps | 1000 |
| save_total_limit | 3 |
| eval_strategy | steps |
| eval_steps | 1000 |
| metric_for_best_model | eval_loss |
| load_best_model_at_end | True |
| dataloader_num_workers | 32 |
| bf16 | True |
| save_safetensors | False |
| gradient_checkpointing | False |

*Table 13.* Dataset used for 1 to M MT model

| Languages | #Train | #Dev | #Test |
|---|---|---|---|
| Hindi−Marathi | ∼ 2M | 997 | 1024 |
| Hindi−Dogri | ∼ 25.2K | 997 | 1024 |
| Hindi−Konkani | ∼96.3K | 997 | 1024 |
| Hindi−Maithili | ∼23.6K | 997 | 1024 |
| Hindi−Nepali | ∼0.12M | 997 | 1024 |
| Hindi−Sanskrit | ∼35.7K | 997 | 1024 |

*Table 14.* Model-based word segmentation results

| Models | hin | | | | mar | | | |
|---|---|---|---|---|---|---|---|---|
| | EM | P | R | F1 | EM | P | R | F1 |
| mT5-Small | 80.820 | 0.977 | 0.972 | 0.972 | 96.71 | 0.994 | 0.994 | 0.994 |
| mT5-Base | 80.76 | 0.9774 | 0.9980 | 0.9725 | 97.084 | 0.9952 | 0.9958 | 0.9951 |
| ByT5-Base | 84.846 | 0.9797 | 0.9821 | 0.9791 | 98.477 | 0.9979 | 0.999 | 0.9983 |

*Table 15.* Machine Translation results on IN22-Gen. BLEU scores are reported for **Hindi to Marathi** and **Marathi to Hindi** translation.

| | HIN → MAR | | MAR → HIN | |
|---|---|---|---|---|
| | 16k | 32k | 16k | 32k |
| BPE | 10.5 | 9.0 | 13.7 | 14.2 |
| Lookup + BPE | 9.6 | 9.6 | 14.1 | 13.3 |
| Model WS + BPE | 9.9 | 9.6 | 14.1 | 13.3 |
| CBPE | 10.3 | 9.8 | 14.4 | 14.3 |
| Lookup + CBPE | 10.0 | 9.6 | 14.2 | 13.9 |
| Model WS + CBPE | 9.9 | 9.3 | 13.5 | 13.9 |

## E. Marathi to Hindi MT correlations with Rényi's efficiency

The Marathi to Hindi MT correlations scores of Rényi's efficiency with chrF2 scores are shown in Table 8.

## F. Perplexity and loss comparison for language modeling

Figure 5 and 6 illustrate the impact of word segmentation for different strategies on language modeling performance in terms of perplexity and loss. It is evident that the lookup-enhanced approaches (`Lookup + BPE` and `Lookup + CBPE`) achieve lower perplexity and loss compared to their standard counterparts (BPE and CBPE). This suggests that leveraging segmented words through lookup-based enhancements helps in better language modeling. Notably, `Lookup + CBPE` achieves the lowest loss and perplexity, reinforcing the idea that segmentation strategies incorporating lookup mechanisms can improve model efficiency.

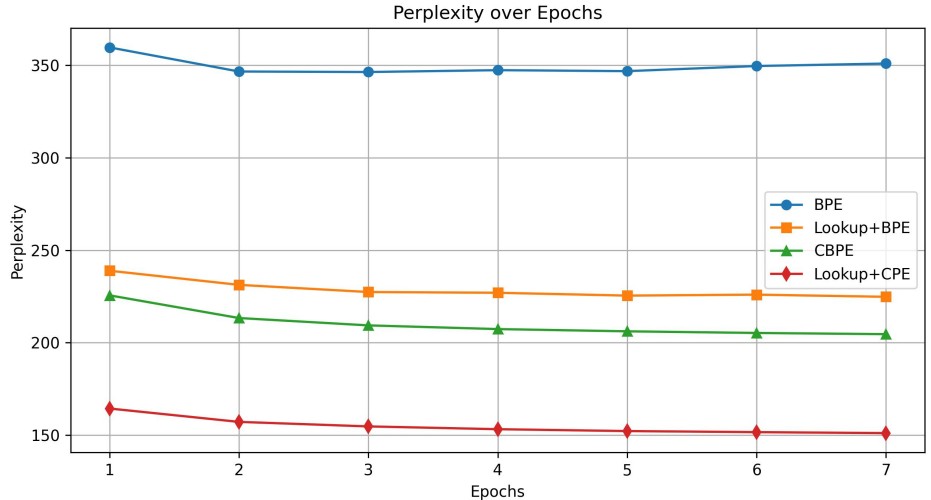

*Figure 5.* Comparison of Perplexity over Epochs

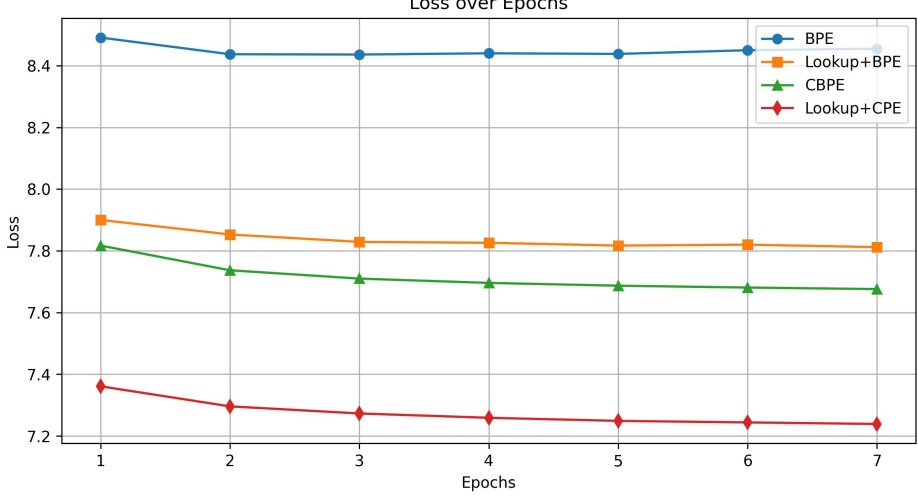

*Figure 6.* Comparison of Loss over Epochs

## G. Multilingual (1 to M) translation analysis

The MT results for Hindi → {Dogri, Konkani, Maithili, Marathi, Nepali, Sanskrit} are shown in Table 16.

*Table 16.* BLEU, chrF2 scores for BPE, Lookup + BPE, CBPE and Lookup + CBPE for Hindi to {Dogri, Konkani, Maithili, Marathi, Nepali, and Sanskrit} MT with 8k, 16k, and 32k merges.

| Method | Metric | doi | | | gom | | | mai | | | mar | | | npi | | | san | | |
|---|---|---|---|---|---|---|---|---|---|---|---|---|---|---|---|---|---|---|---|
| | | 8k | 16k | 32k | 8k | 16k | 32k | 8k | 16k | 32k | 8k | 16k | 32k | 8k | 16k | 32k | 8k | 16k | 32k |
| **BPE** | **BLEU** | 21.6 | 21.3 | 21.4 | 11.4 | 11.2 | 12.1 | 13.9 | 14.0 | 13.6 | 9.2 | 9.5 | 9.8 | 10.1 | 10.1 | 9.9 | 8 | 8.2 | 7.7 |
| | **chrF2** | 49.0 | 48.9 | 48.8 | 41.0 | 41.0 | 40.7 | 46.6 | 46.6 | 45.8 | 40.6 | 40.2 | 39.9 | 44.7 | 44.6 | 44.6 | 35.9 | 35.8 | 35.4 |
| **Lookup + BPE** | **BLEU** | 21.5 | 21.4 | 21.1 | 10.3 | 11.9 | 11.7 | 13.7 | 14.4 | 13.6 | 9.0 | 9.9 | 9.8 | 9.7 | 9.8 | 10.0 | 7.6 | 8.1 | 7.9 |
| | **chrF2** | 48.8 | 48.9 | 48.5 | 40.3 | 41.1 | 41 | 46.3 | 46.4 | 45.7 | 40.1 | 40.5 | 39.8 | 44.4 | 44.5 | 44.5 | 35.2 | 35.9 | 35.6 |
| **CBPE** | **BLEU** | 21.5 | 21.6 | 21.3 | 12.1 | 11.4 | 12.8 | 14.1 | 13.8 | 13.9 | 9.4 | 9.6 | 10.6 | 10.3 | 10.0 | 9.4 | 7.9 | 7.6 | 7.3 |
| | **chrF2** | 49.1 | 49.0 | 48.5 | 41.0 | 40.9 | 40.6 | 46.5 | 46.1 | 45.4 | 39.8 | 40.2 | 39.8 | 44.8 | 44.6 | 43.7 | 35.9 | 35.5 | 34.8 |
| **Lookup + CBPE** | **BLEU** | 21.4 | 21.3 | 20.9 | 11.6 | 11.6 | 12.1 | 14.1 | 13.2 | 14.0 | 9.7 | 9.3 | 10.1 | 9.9 | 10.1 | 9.8 | 7.7 | 7.4 | 7.2 |
| | **chrF2** | 48.8 | 48.6 | 48.2 | 41.1 | 40.5 | 40.5 | 46.7 | 45.7 | 45.5 | 40.8 | 39.2 | 40.0 | 44.8 | 44.4 | 44.3 | 36.0 | 35.0 | 34.6 |

| Hindi Word | BPE Segmentation (32k) | SCORE | Lookup+BPE Segmentation (32k) | SCORE |
|---|---|---|---|---|
| अंतरा | अंतर@@ ा | 4 | अंतर@@ ** ा | 4 |
| अजैविक | अ@@ जैविक | 4 | अ@@ ** जैविक | 4 |
| अपचयन | अप@@ चयन | 4 | अप@@ ** चयन | 4 |
| अर्थपूर्ण | अर्थ@@ पूर्ण | 4 | अर्थ@@ ** पूर्ण | 4 |
| अश्विनीकुमार | अश्@@ वि@@ नी @@ कुमार | 2 | अश्@@ वि@@ नी@@ ** कुमार | 2 |
| अष्टावक्र | अ@@ ष्टा@@ व@@ क्र | 1 | अ@@ ष्@@ टा** व@@ क्र | 1 |
| असताना | अस@@ ताना | 4 | अस** ताना | 4 |
| आगरकर | आग@@ रकर | 1 | आग@@ र** कर | 1 |
| आठवले | आठवले | 4 | आठव** ले | 4 |
| आनंददायी | आनंद@@ दायी | 4 | आनंद@@ ** दायी | 4 |
| आश्चर्यजनक | आश्चर्यजनक | 4 | आश्चर्य** जनक | 4 |
| उतरता | उतरता | 4 | उतर** ता | 4 |
| उतरते | उतरते | 4 | उतर** ते | 4 |
| उतरवा | उतर@@ वा | 4 | उतर@@ व** ा | 2 |
| **उद्वहन** | **उ@@ द्व@@ हन** | **1** | **उद्@@ ** वहन** | **4** |
| **उपजता** | **उप@@ जता** | **1** | **उपज** ता** | **4** |
| **उपजेल** | **उप@@ जेल** | **1** | **उपज** े ल** | **4** |
| उपनगर | उपनगर | 4 | उप** नगर | 4 |
| उभारता | उभारता | 4 | उभार** ता | 4 |
| उभारते | उभार@@ ते | 4 | उभार** ते | 4 |
| उभारा | उभारा | 4 | उभार** ा | 4 |
| उभारे | उभारे | 4 | उभार** े | 4 |
| एकरूपता | एकरूपता | 4 | एक** रूपता | 4 |
| ऑस्ट्रेलियाने | ऑ@@ स्ट्रे@@ लिया@@ ने | 2 | ऑ@@ स्ट्रे@@ लिया@@ ** ने | 2 |
| **करकरे** | **कर@@ करे** | **2** | **कर@@ कर** े** | **4** |
| **कल्पता** | **कल्@@ पता** | **1** | **कल्@@ प** ता** | **2** |
| **कल्पा** | **कल्@@ पा** | **1** | **कल्@@ प** ा** | **2** |
| **कांडला** | **का@@ ंड@@ ला** | **1** | **कांड** ला** | **4** |
| **कांडा** | **का@@ ंडा** | **1** | **कांड** ा** | **4** |
| **काकडे** | **का@@ क@@ डे** | **1** | **का@@ क@@ ड** े** | **2** |
| **काटता** | **का@@ टता** | **1** | **काट** ता** | **4** |
| काटते | काटते | 4 | काट** ते | 4 |
| **कातते** | **का@@ तते** | **1** | **का@@ त** ते** | **3** |
| कातरू | का@@ तर@@ ू | 2 | का@@ तर@@ ** ू | 2 |
| **कापता** | **का@@ पता** | **1** | **का@@ प** ता** | **3** |
| कार्यकर्ता | कार्यकर्ता | 4 | कार्य** कर्ता | 4 |
| कालखंड | कालखंड | 4 | काल** खंड | 4 |
| किरकिरा | किरकि@@ रा | 1 | किरकि@@ र** ा | 1 |
| किरकिरे | किरकि@@ रे | 1 | किरकि@@ र** े | 1 |
| कुरकुरा | कुर@@ कु@@ रा | 1 | कुर@@ कु@@ र** ा | 1 |
| कुरकुरे | कुर@@ कु@@ रे | 2 | कुर@@ कु@@ र** े | 2 |
| कोंडली | को@@ ंड@@ ली | 2 | को@@ ंड@@ ** ली | 2 |
| कोंडा | कोंडा | 4 | को@@ ंड@@ ** ा | 2 |
| कोंबो | को@@ ंब@@ ो | 1 | को@@ ंब@@ ** ो | 1 |
| क्रमवार | क्रम@@ वार | 4 | क्र@@ म** वार | 2 |
| खर्चा | खर्चा | 4 | खर्च** ा | 4 |

*Figure 7.* Sample EvalTok scores for BPE and Lookup + BPE segmentation.

