# OpenReview forum: "MorphTok: Morphologically Grounded Tokenization for Indic languages"
_ICML.cc/2025/Workshop/TokShop — TokShop_

### Official Review · Reviewer_cL7v · 2025-06-05
**Good work, but more linguistic details would improve the clarity of the message**

**Rating:** 7
**Confidence:** 2

**Review:**

The paper presents a novel approach to tokenization specifically designed for Indic languages. The approach based on a morphological pre-tokenization, pre-splitting words into morphemes, and also normalizing the morphemes to match existing words (if I understand this correctly). The authors evaluate the approach both automatically intrinsically in LM and MT, showing improvements in language model perplexity as well as machine translation quality, as well es manually extrinsically (using a novel proposed evaluation approach), confirming that their method indeed leads to more correct tokenization. They also release a morphosegmentation dataset which they created. In general, this is a wide range of valuable contributions, with a well motivated approach that is shown to be useful in practice.

For a reader without a good knowledge of Indic languages, I would like the paper to better explain the specifics of the Indic languages. From my understanding, the first step of morpho-splitting is rather standard, but hindered by the fact that a morpho-segmented dataset had not been available for the target languages. But then there is the step which I understand as a normalization of some changes on the morpheme boundaries, so that the produced morphemes are identical to existing words (whereas as parts of a complex word there are some changes at the morpheme boundary). I would need this part to be explained in more detail, what kind of changes there are, how regular they are, if it is possible to restore the original word using rules, or at least vice versa if it is possible to do the changes in a rule-based way (which could be used to generate training data)...

My other remark is that the related work is surprisingly brief, as if morphological splitting had basically never been done before. In such a paper, I would expect some discussion of semi-supervised and unsupervised morphological splitting methods, previous attempts at using morphology-based tokenization or pretokenization, as well as some discussion of preexisting approaches for morpheme normalization. If nothing is published for Indic languages, I still am quite sure that there is some work e.g. for splitting of German compound nouns, which also undergo some changes at the morpheme boundaries, which might or might not be similar to what happens in Indic languages, and orthogonally the approaches that work for German might or might not be applicable to Indic languages; that I do not know, but that is the task of the authors of such a paper, to seek and study existing related work and to discuss its (non-)relevanci in their paper.

---

### Official Review · Reviewer_4egC · 2025-06-06
**The paper proposes a pre-tokenisation method for BPE, a new tokenisation method that improves BPE for Inidic languages, and a human evaluation metric for tokenisation.**

**Rating:** 5
**Confidence:** 3

**Review:**

The paper proposes a pre-tokenization method for BPE (based on look-up tables or language models), a new tokenization method based on BPE (CBPE) that does not separate the dependent vowels, and a human evaluation metric for tokenization.

**Strengths**:

- Both the pre-tokenisation method and CBPE do not hurt the downstream task's performance while producing better tokens from a linguistic point of view.
- The segmentation dataset is useful for future research.

**Weaknesses**:

- The lookup table requires human annotations and expertise.
- The improvement in fertility is minimal in most cases (e.g., 1.8858 vs 1.8174 is roughly 7 fewer tokens each 100 words)
- For EvalTok: How many human annotators? Was each sample annotated by just one annotator or multiple annotators?
- Are the 100 words sampled for EvalTok from the lookup table?
- Sometimes, it is not clear exactly which setting or model is being evaluated. For example, on line 379 in the “Human Evaluation of Tokenization,” does “pre-tokenization+BPE” use the lookup table or the language model?
- Table 7: I do not think this is a fair comparison because the other models treat all the languages and scripts the same way, while CBPE is explicitly designed not to split the dependent vowel.
- It is not clear what is described in Section 6.4. A shorter segment size seems to suggest a larger number of tokens, so higher fertility. Also, were the words sampled from In22-Gen or from other sources? Why a sample of 395 words and not, for example, the entire test set?
- It would be interesting to see the fertility of Lookup+BPE, even though it does not reflect the linguistic validity of token splits. It still gives information on how efficient it would be to run models with this tokeniser.

**Minor issues**:
- On lines 378 and 372, “Human Evaluation of MT Results” and “Human Evaluation of Tokenization”: these should be formatted as subsections or paragraphs
- Table 15 should be in the main text and not in the appendix, since it is mentioned on line 424
- There are multiple repetitions in the paper. For example, EvalTok is introduced in Section 4.2, and then in Section 5.2, line 374, “we employ the EvalTok metric, detailed in Section 4.2,” and line 77, "we propose a new metric EvalTok, as described in Section 4.2, line 398, “EvalTok metric as described in Section 4.2.”

---

### Official Review · Reviewer_oD9w · 2025-06-08
**Decent improvement from BPE for Indic languages**

**Rating:** 7
**Confidence:** 3

**Review:**

This paper provides the following:
- A linguistically informed segmentation method that aligns subword units with morpheme boundaries, improving linguistic coherence compared to standard BPE.
- An extension of BPE that prevents dependent vowel diacritics from forming separate tokens, thereby reducing token fragmentation in abugida scripts and enhancing compression, while maintaining comparable or improved downstream performance.
- A human-centric evaluation metric designed to assess the morphological and semantic quality of tokenization, providing a human-grounded assessment that complements automated metrics.
- Creation and release of a curated morphological segmentation dataset for Hindi (54k entries) and Marathi (58k entries) to support research in morphology-aware tokenization.
- Experiments demonstrate that morphologically grounded tokenization improves machine translation and language modeling performance.  Specifically, Lookup + BPE and Lookup + CBPE configurations consistently outperform standard BPE and CBPE in language modeling by achieving lower perplexity and loss values.

Strengths:

- Addresses a crucial problem for Indic languages: The paper tackles the known issue of BPE's inadequacy for morphologically rich languages like Hindi and Marathi, particularly concerning linguistically incoherent segmentations and dependent vowels.
- Novel Contributions: The introduction of morphology-aware pre-tokenization, Constrained BPE (CBPE), and the EvalTok human evaluation metric are significant contributions to the field.
- Comprehensive Evaluation: The methods are rigorously evaluated using both automatic metrics (BLEU, chrF, COMET, perplexity, loss, fertility) and a new human evaluation metric (EvalTok) on downstream tasks like machine translation and language modeling.
- Dataset Release: The creation and release of a specialized dataset for Hindi and Marathi segmentation is valuable for future research.
- Practical Relevance: The proposed methods aim to improve the efficiency and performance of LLMs for Indic languages, which has significant practical implications.

Weakness

- The paper was evaluated only on two languages Hindi and Marathi, I need more evaluation is needed.
- I would have also liked to see the translation from English to other Indic languages in the evaluation
- Specific to Indic Languages/Abugida Scripts: While a strength in its focus, the direct applicability of all proposed constraints (like dependent vowel handling) is primarily for abugida writing systems and may not generalize directly to all language families without adaptation.

---

### Decision · Program_Chairs · 2025-06-10

Accept